# Climate Change and Diarrhoeal Disease Burdens in the Gaza Strip, Palestine: Health Impacts of 1.5 °C and 2 °C Global Warming Scenarios

**DOI:** 10.3390/ijerph19084898

**Published:** 2022-04-18

**Authors:** Shakoor Hajat, David Gampe, Amal Sarsour, Samer Abuzerr

**Affiliations:** 1Centre on Climate Change and Planetary Health, London School of Hygiene & Tropical Medicine, London WC1E 7HT, UK; 2Department of Geography, Ludwig Maximilian University, 80333 Munich, Germany; d.gampe@lmu.de; 3United Nations University—Institute on Comparative Regional Integration Studies (UNU-CRIS), 8000 Bruges, Belgium; asarsour@cris.unu.edu; 4Environmental Health Unit, Gaza Ministry of Health, Gaza P860, Palestine; samer_516@hotmail.com; 5Department of Social and Preventive Medicine, School of Public Health, University of Montreal, Montreal, QC H3N 1X9, Canada

**Keywords:** climate change, ambient temperature, rainfall, diarrhoea, time-series regression, risk assessment, Gaza

## Abstract

The Gaza Strip is one of the world’s most fragile states and faces substantial public health and development challenges. Climate change is intensifying existing environmental problems, including increased water stress. We provide the first published assessment of climate impacts on diarrhoeal disease in Gaza and project future health burdens under climate change scenarios. Over 1 million acute diarrhoea cases presenting to health facilities during 2009–2020 were linked to weekly temperature and rainfall data and associations assessed using time-series regression analysis employing distributed lag non-linear models (DLNMs). Models were applied to climate projections to estimate future burdens of diarrhoeal disease under 2 °C and 1.5 °C global warming scenarios. There was a significantly raised risk of diarrhoeal disease associated with both mean weekly temperature above 19 °C and total weekly rainfall below 6 mm in children 0–3 years. A heat effect was also present in subjects aged > 3 years. Annual diarrhoea cases attributable to heat and low rainfall was 2209.0 and 4070.3, respectively, in 0–3-year-olds. In both age-groups, heat-related cases could rise by over 10% under a 2 °C global warming level compared to baseline, but would be limited to below 2% under a 1.5 °C scenario. Mean rises of 0.9% and 2.7% in diarrhoea cases associated with reduced rainfall are projected for the 1.5 °C and 2 °C scenarios, respectively, in 0–3-year-olds. Climate change impacts will add to the considerable development challenges already faced by the people of Gaza. Substantial health gains could be achieved if global warming is limited to 1.5 °C.

## 1. Introduction

Climate change is now increasingly recognised as a fundamental threat to human health and wellbeing, especially in low- and middle-income (LMI) countries that have the least economic capacity to adapt to such changes. Higher than average temperature increases are projected for the Middle East, which already experiences some of the hottest temperatures on the planet and is the world’s most water-stressed region [1]. Some of the biggest challenges are faced by the population of Gaza in the Occupied Palestinian Territories which has been classified by the Organization for Economic Co-operation and Development as a highly fragile region. This is due to its severe rates of poverty, poor infrastructure services, high population density, and political conditions.

As temperatures continue to rise and precipitation decreases, the risk of drought in the Gaza Strip will increase [2]. This will intensify existing development challenges for the Gaza Strip, including increasing food insecurity and water stress. The only natural water resource in Gaza is the Coastal Aquifer Basin [3], however the quantity and quality of its freshwater is already insufficient to meet domestic needs [4]. The aquifer has been degraded by overextraction, sewage contamination, and seawater intrusion [5]. Already today, over 90% of water resources in Gaza are undrinkable. Children’s access to clean water in Gaza decreased from 98% in 2000 to 11% in 2014, whereas the rate remained stable in the West Bank [6]. Consequently, access to safe drinking water, a basic human right, is becoming increasingly difficult and many public health indicators are worsening as a result [7]. Prominent among these are the very high rates of WASH-related diseases in Gaza, including diarrhoeal disease.

Diarrhoeal disease is the second leading cause of death in children under 5 worldwide and the second leading cause of death and disability in LMI countries [8]. High baseline disease rates mean that even small changes in risk associated with climate factors have the potential to substantially add to public health burdens [9]. High temperatures, heavy rainfall, and droughts have all been implicated in raising the risk of diarrhoea cases in selected settings; however, there is very little evidence from Gaza despite the high disease prevalence. One study observed seasonality in cryptosporidiosis in Gaza children, with higher rates in the hot dry season [10]. However, to the best of our knowledge, no published studies have assessed the impacts of climate factors on diarrhoeal disease in Gaza.

Furthermore, no studies have projected future health burdens in Gaza under climate change scenarios, despite the unique set of environmental, socio-economic, and political challenges that the population faces. Many countries have reaffirmed their commitment to achieving the Paris agreement (UNFCCC 2015) goals of “holding the increase in the global average temperature to well below 2 °C above pre-industrial levels and pursuing efforts to limit the temperature increases to 1.5 °C above pre-industrial levels”. Estimating the differential health impacts associated with these two Global Warming Levels (GWLs) have the advantage over health burdens projected for arbitrary future time horizons since the periods when such levels are exceeded differ considerably between climate models, therefore also strongly impacting regional warming. Consequently, communicating the impacts of climate change based on GWLs have more relevance for policymakers compared to using predefined time periods. Therefore, there is a greater need to demonstrate the public health gains to be made from achieving the more stringent 1.5 °C target, especially for populations where this evidence is currently lacking.

For the first time, we assess the impacts of climate factors on acute diarrhoeal disease in Gaza using weekly time-series datasets and undertake a health impact assessment to quantify future burdens under climate change scenarios.

## 2. Materials and Methods

### 2.1. Study Setting

The Gaza Strip in the Occupied Palestinian Territories is a territory in the Middle East with a surface area of 365 square Kilometres along the Mediterranean Sea and northeast of Egypt (Figure 1). The estimated population of the Gaza Strip is 2,136,507 million [11]. About 64% of the Gaza population are refugees and 44.4% of the refugee population reside in eight densely crowded refugee camps.

### 2.2. Health Data

Data on the weekly number of acute diarrhoea cases presenting to health facilities in Gaza between 2009–2020 were obtained from the records of the epidemiology department at the Palestinian Ministry of Health in the Gaza Strip. The data come from all health providers and facilities in the Gaza Strip participating in the communicable diseases surveillance system. There are four types of healthcare providers in Gaza: the Government, the United Nations Relief and Works Agency (UNRWA), NGOs, and the private sector. The Government and UNRWA offer health services to most of the residents of Gaza, with NGOs and the private sector serving smaller sections of the population.

In the Gaza Strip, a disease-specific approach to communicable disease surveillance is applied, which is dependent on a passive surveillance system. Of the 161 registered health facilities in Gaza, 112 (69.5%) participate in the surveillance and reporting of communicable diseases. The main health providers who participate in diarrhoeal disease surveillance are from primary health care centres who constituted 60% of all providers in our study. The remainder of the notifications were from hospitals and from Government and NGO laboratories where diagnosis of diarrhoea is primarily based on stool analysis.

### 2.3. Climate Data

Current climate data are represented by the ERA-5 Land reanalysis dataset obtained through the Copernicus Climate Change Service (C3S) [12]. ERA-5 Land provides gridded information on precipitation and air temperature, among other variables, at a 9 km spatial resolution. This dataset was selected as local station data were not available for the study area. Gridded observational datasets lack weather stations within the Gaza Strip and so can only estimate meteorological conditions through interpolation from stations in surrounding countries. To correspond with the health data, the reference period of the climatological data was set to 2009–2020. The datasets show annual average temperatures of 20.98 °C and 303 mm annual precipitation over the reference period.

Projections of future climate were obtained from an ensemble of regional climate models (RCMs) provided through the Coordinated Regional Climate Downscaling Experiment (EURO-CORDEX, http://www.cordex.org/, accessed on 9 December 2021) [13]. The RCM ensemble applied in this study comprises of 13 models which provide daily precipitation and temperature on a 0.11° grid (roughly 12.5 km). These are summarised in Table 1. Datasets were bias-adjusted using a distribution based scaling approach by the Swedish Meteorological and Hydrological Institute, Rossby Centre, and are publicly available at the Earth System Grid Federation (ESGF). Following the temporal resolution of the health data, all climate data were aggregated to weekly averages (temperature) or sums (precipitation).

We project future diarrhoeal disease burdens based on the GWL targets of 1.5 °C and 2.0 °C. Therefore, in contrast to using predefined time horizons, our future periods are defined by the time global temperatures are expected to exceed the GWLs of 1.5 °C and 2.0 °C. Here, a 20-year average period centred around the year when the two GWLs are projected to be exceeded are applied and obtained from Hauser et al. [14]. For the RCMs applied, the corresponding periods when targets are projected to be exceeded span the years 2007–2047 for 1.5 °C and 2024–2068 for 2 °C. This approach avoids additional uncertainties originating from different scenarios (e.g., RCP 4.5 vs. 8.5). Provided a physical link between increased GWLs and expected changes in hot/dry extremes, the selection of a different scenario would thus only shift the time horizon when GWLs are reached.

### 2.4. Analysis

#### 2.4.1. Epidemiologic Analysis to Model Current Impacts

Poisson time-series regression models were used to assess acute associations between weekly diarrhoea cases and mean weekly ambient temperature and total weekly rainfall. A series of cubic-spline functions of time were used to flexibly model underlying trends and seasonal patterns in the health data unrelated to climate factors. Seasonally-adjusted models used a scale parameter to allow for overdispersion and model residuals were lagged by one week to adjust for significant residual autocorrelation associated with the week before.

Next, distributed lag non-linear models (DLNMs) employing cross-basis functions were used to model any delayed and non-linear effects of each exposure, including identification of thresholds in the relationships at which risk of diarrhoeal disease increases [15]. This allows for the relationship to be simultaneously modelled at different lags of exposure. A maximum lag of 4 weeks was used to capture delayed effects. These models suggested an increased linear risk associated with high temperatures at lag 0–1 weeks and with low rainfall at lags 0–4 weeks. The cumulative effects of temperature or rainfall across all important lags were therefore estimated using distributed lag linear threshold models above the identified threshold in the case of temperature, and below the threshold for rainfall. Each climate variable was investigated separately, but final quantification of estimates was based on a model with both factors simultaneously entered, i.e.,:Log[E(Y*_i_*)] = α + β_1_T*_i,l_* + β_2_R*_i,l_* + β_3_ns(time) + γr*_i,l_*
where E[Y*_i_*] is expected diarrhoea cases on week *i*; T*_i_**_,l_* is the distributed lag of mean weekly temperature > 19 °C and lag 1 week; R*_i,l_* is the distributed lag of total weekly rainfall < 6 mm and lag 4 weeks; ns = natural spline functions of time; and r*_i,l_* is deviance residuals at 1 week lag obtained from the model without any autoregressive terms. A separate indicator term for hot dry weeks, i.e., weeks both above the heat threshold and below the rainfall threshold, was also modelled but additional risk on such weeks was not found to be present.

#### 2.4.2. Health Impact Assessment to Model Future Burdens

The climate-health relationships characterised above were then applied to climate projections data to estimate future numbers of diarrhoeal disease cases under climate change scenarios. The risk function is assumed to remain unchanged in future periods, indicating no reduced risk in future due to adaptation or, conversely, no increased vulnerability. This may be an unrealistic scenario, but our aim was to characterise the potential changes associated with climate factors only. The excess number of diarrhoea cases attributable to higher temperatures or lower rainfall compared to the baseline period (2009–2020) were estimated for each climate model under each emission scenario using the following formula:$$ED=\overset{N}{\underset{i=1}{\sum }}BD_{i}\left(\frac{RR_{i}-1}{RR_{i}}\right)$$
where *ED* is the total number of excess diarrhoea cases attributable to the climate factor (temperature or rainfall) over time-period *N*, *BD_i_* is the baseline diarrhoea cases in week *i*, and *RR_i_* is the relative risk of diarrhoea cases for week *i* associated with a unit change in the climate factor above/below the identified threshold. For each climate model and GWL, projections are averaged across all years within each 20-year period described in Table 1.

Analyses were conducted in STATA 17.0 (www.stata.com). The study protocol was approved by the Palestinian Health Research Council (Helsinki Committee for Ethical Approval research number: PHRC/HC/1036/22) and the Palestinian Ministry of Health.

## 3. Results

### 3.1. Descriptive Results

During the 2009–2020 study period, there were a total of 706,699 and 392,499 diarrhoea cases presenting to participating health facilities among subjects aged 0–3 years and greater than 3 years, respectively. Figure 2 shows the annual distribution of weekly diarrhoea cases and average weekly mean temperature (°C) and total weekly rainfall (mm) in Gaza averaged across years 2009–2020. Diarrhoea cases peak in early summer with the seasonality being most pronounced in those aged 0–3 years. On average, there were 31.3 weeks per year when mean weekly temperature was above the threshold of 19 °C and 32.8 weeks per year when total weekly rainfall was below 6 mm.

### 3.2. Current Impacts

Figure 3 shows the seasonally-adjusted relationship between (a) temperature and (b) rainfall with the relative risk of diarrhoea presentation in 0–3-year-olds across multiple lags based on the DLNM model. There was a linearly increasing diarrhoea risk associated with high temperatures at lags 0 and 1 weeks, but no effect at longer lags. The heat threshold was estimated at 19 °C based on model deviance statistics. The relationship with rainfall exhibited a broadly linear increased diarrhoea risk associated with low rainfall, with an identified threshold at 6 mm and the effect persisting for up to 4 weeks following exposure. Relationships in the >3 years age-group were similar but not as strong (not shown). We observed no increased risk of diarrhoea associated with high rainfall. Table 2 quantifies exposure effects based on linear-threshold models and the resulting annual number of diarrhoea cases attributable to high temperature and low rainfall. Given the high baseline rates of diarrhoea, a substantial number of cases among 0–3 years can be attributed to heat and to low rainfall in particular.

### 3.3. Future Burdens

Figure 4 shows the average number of hot (temperature > 19 °C) and dry (rainfall < 6 mm) weeks per year in Gaza projected under current and future climate. For both future warming levels, the applied climate models project an increase in both hot and dry weeks. The number of weeks above the temperature threshold rises considerably under a 2 °C warming scenario. For the baseline period, the RCMs show a slight overestimation in hot and dry weeks compared to ERA-5. However, the selected baseline period (2009–2020) overlaps with the time period of 1.5 °C GWL for some models (Table 1). Therefore, these models already show an increased warming with respect to the observed temperature increase (around 1 °C). Consequently, the number of hot and dry weeks is slightly altered compared to ERA-5. This should not be interpreted as model bias and does not affect the future burdens projected in this study.

We project an annual average of 2446 heat attributable cases in the 0–3-year age-group under a 2 °C global warming level, which amounts to a 10.7% increase from modelled baseline levels (reference period 2009–2020). However, this increase would be limited to only 1.8% under a 1.5 °C scenario. For the >3-year age-group, rises of 1.8% and 10.9% are projected under the 1.5 °C and 2 °C scenarios, respectively, from a much lower baseline. The boxplots in Figure 5 show the distributions of annual number of heat attributable diarrhoea cases across the 13 climate models at baseline and for each warming level. The distributions of projections across the climate models were highly negatively skewed.

Diarrhoea increases associated with reduced rainfall under the two warming levels are smaller, but from a much higher baseline. For the 0–3 age-group, mean rises of 0.9% and 2.7% are projected for the 1.5 °C and 2 °C scenarios, respectively. This was not quantified for the >3 years age-group as the rainfall effect was not statistically significant. Figure 6 shows the distribution of annual cases in the 0–3 age-group across the 13 climate models.

## 4. Discussion

Our results reveal a considerable current burden of acute diarrhoeal disease in Gaza associated with low rainfall and high temperatures, which will further increase as the climate continues to change. We demonstrate that heat-related diarrhoeal disease could rise by over 10% under a 2 °C global warming level, but that this would be limited to under 2% if the 1.5 °C target could be achieved instead. Although such longer-term impacts will need to be anticipated, adaptation measures will be a necessary complement to mitigation to safeguard against the impacts expected in the near-term regardless of mitigation actions taken. Indeed, our results suggest that much preventative action needs to be taken now to improve the existing situation.

Diarrhoea incidence in children has decreased over time in many parts of the world [8]; however, the burden in Gaza remains high. A survey of children under 5 in Gaza reported that 14% had experienced an episode of diarrhoea in the 2 weeks before interview [16]. Many children under 5 are admitted to hospital because of diarrhoea, with low family income, urban residence, and use of municipal water for drinking and cooking all increasing risk [17,18,19]. This reflects the dire and worsening issues of water scarcity and food security in Gaza which climate change will further exacerbate. The great majority of food is imported into the area which makes it highly vulnerable to future climatic, economic and political shocks.

The risk of diarrheal disease we observed with high temperatures in Gaza was smaller than that reported in many other settings [20]; however, heat-attributable numbers were still large due to the high baseline rates of disease coupled with the fact that temperatures were not particularly extreme before adverse effects became apparent. The effects of temperature exposure on disease were mostly immediate, which suggests an acute mechanism of association, such as increased consumption of contaminated water or increased food spoilage during hot weather; this could inform potential behavioural changes whenever high temperatures are forecast. However, climate change may also lead to temperature impacts via other pathways, such as extending the transmission season leading to increased survival of pathogens outside of the host [21]. Diarrhoea risk was greatest in those aged 0–3 years, which may be more closely linked to Cryptosporidiosis compared to older ages [22].

Low rainfall was the climate factor most strongly related to diarrhoea risk in our study. Drought conditions can increase pathogen concentrations and encourage the use of water sources of poorer quality [23]. Although water stress is an increasing concern for many countries in the Middle East region, the problem of clean water access in Gaza is most urgent. Under various population growth and water availability scenarios, Feitelson et al. conclude that the Gaza Strip requires augmentation of natural water sources regardless of climate change [4]. They identify desalination as the only feasible option, but such measures are very energy-intensive and the ability to build such a plant in Gaza is highly dependent on the cooperation of neighbouring countries and the relaxation of Israel’s border controls [24]. The role of the private sector in supplying water during emergencies may also become more important [7]. Prompted by the COVID crises, Negev et al. highlight the need for greater regional and intersectoral collaboration to improve resilience in the face of climate change and public health challenges in the Middle East [25]. However, the framing of climate change and water issues as a security risk in the region may impede any progress towards major adaptive measures [26].

Our study has a number of strengths and limitations. Aside from a previous Masters thesis [27], our study is the first to provide estimates of climate impacts on health outcomes in Gaza, and also the first to project health burdens under climate change scenarios in this highly vulnerable setting. We obtained health data over an extended period of time, resulting in over a million cases of diarrhoeal illness that we were able to link to climate information. However, the situation of climate data scarcity, in particular regarding highly heterogeneous variables such as precipitation, remains a challenge in the study area [2]. Furthermore, climate models are prone to biases, especially at the regional scale and for precipitation in particular. Whilst the use of bias-adjusted models reduced this limitation considerably for current climate conditions, limitations remain for future projections [28]. Nevertheless, there is strong agreement in a decreasing trend of precipitation over the Middle East region under global warming [29]. In addition, some of the climate models applied reach the global warming level of 1.5 °C already by 2022, resulting in some overlap with the considered baseline period (2009–2020). Therefore, the projected differences in precipitation and temperature are reduced for these models as they overestimate the current, real world global warming of around 1.0°. Consequently, the projected future burdens may be underestimated in this study. However, this affects both considered scenarios equally and the observation of higher burdens under 2.0° remains.

In modelling our future burdens, we have assumed that the current relationship between weather factors and diarrhoea risk will remain constant in future, and we did not aim to model other factors that will likely also contribute to future burdens since we sought to quantify the contribution from climate change alone. The Gaza Strip is already one of the most densely populated areas of the world and further population growth will amplify future diarrhoea burdens. Incorporation of WASH factors and other socio-economic and ecological indicators into projections could also help to better understand the links between climatic factors and diarrheal disease and inform adaptation strategies [9]; however, the impacts that such development factors will likely have on public health in Gaza cannot viably be predicted in this politically unstable and fragile State. Our results also only provide lower bound estimates of total diarrhoea burdens for the population of Gaza since we have not quantified mortality impacts. In addition, although our health data are sourced from Government, UNRWA, NGO, and private providers, and are, therefore, considered highly representative, they do not provide complete coverage of all healthcare facilities in the State. Finally, we were also unable to subdivide diarrhoea cases by pathogen type which is likely to modify sensitivity to climate factors [20].

## 5. Conclusions

Climate factors contribute to the high rates of diarrhoeal disease currently observed in Gaza. It is a population that has experienced both forced mobility and now forced immobility, which both have the potential to heighten vulnerability to climate change processes [30]. Our assessment shows that the degree of global warming expected will have a large bearing on future diarrhoea burdens in the region and provides further motivation for the world to achieve more stringent mitigation targets. Although climate change is not the most immediate concern for the Gaza Strip, future climate-related impacts will add to the considerable public health and development challenges already faced by the people of Gaza.

## Figures and Tables

**Figure 1 ijerph-19-04898-f001:**
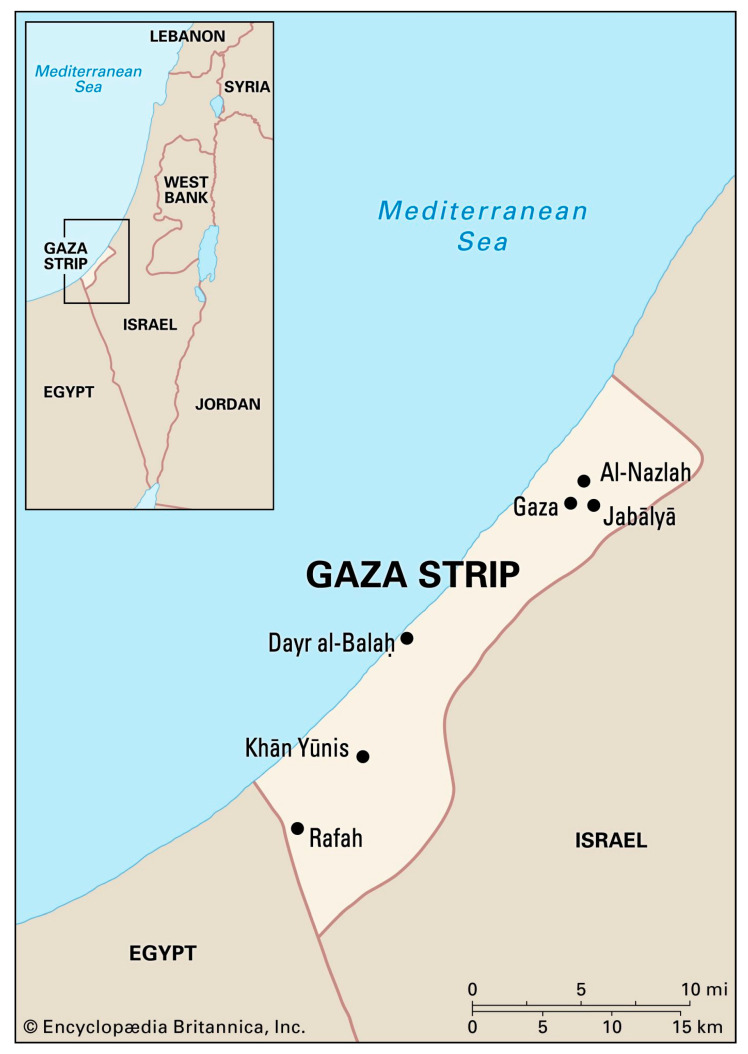
Map of the Gaza Strip.

**Figure 2 ijerph-19-04898-f002:**
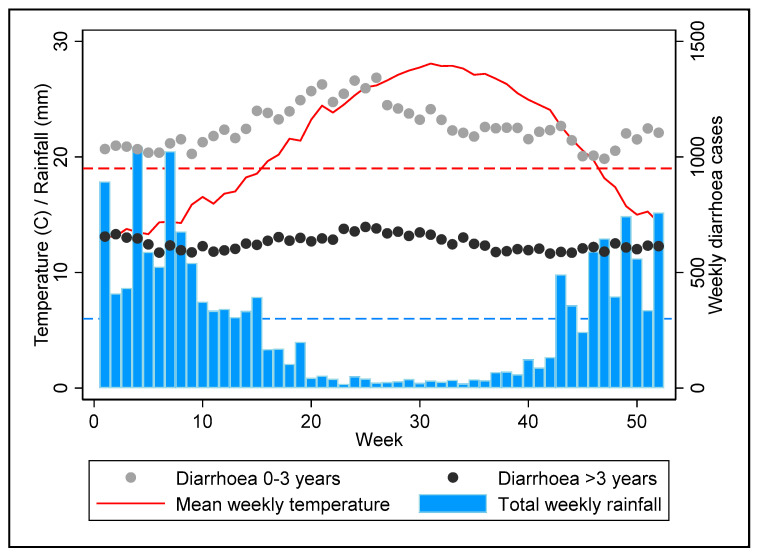
Annual distribution of weekly diarrhoea cases and climate factors across years 2009–2020. The dashed red and blue lines denote threshold values for temperature (>19 °C) and rainfall (<6 mm) respectively.

**Figure 3 ijerph-19-04898-f003:**
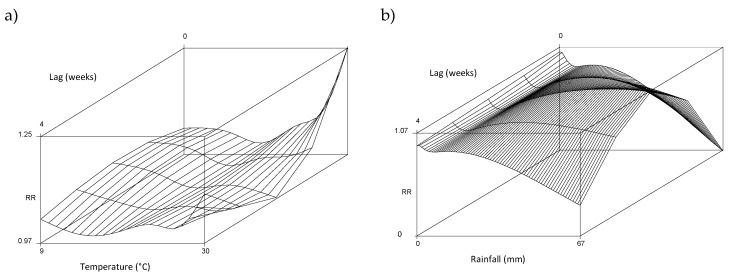
Seasonally adjusted relationship between (**a**) temperature and (**b**) rainfall with relative risk of diarrhoea presentation in the 0–3-year age-group.

**Figure 4 ijerph-19-04898-f004:**
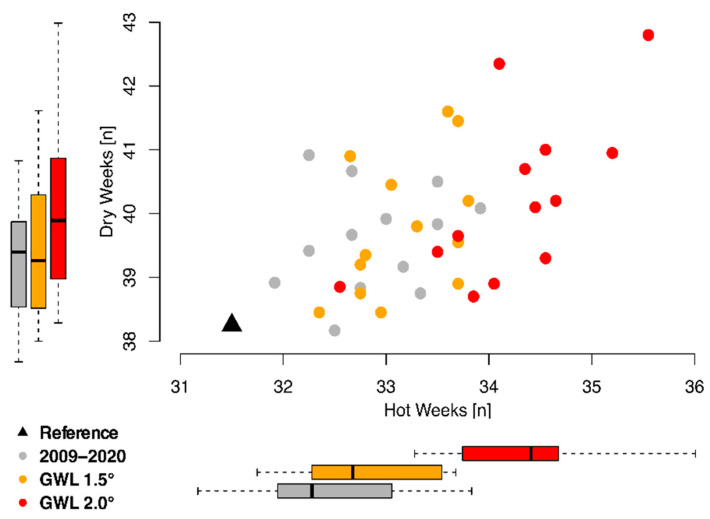
Annual average number of hot (temperature > 19 °C) and dry (rainfall < 6 mm) weeks under current and future climate. The corresponding number of weeks are shown for the reference data (ERA5, black triangle), as well as the applied climate models for the reference period 2009–2020 (grey) and the two Global Warming Level (GWL) targets of 1.5 °C (orange) and 2.0 °C (red). Outside boxplots refer to the spread of the climate models for hot and dry weeks (x and y-axis) separately under the different time periods/warming levels.

**Figure 5 ijerph-19-04898-f005:**
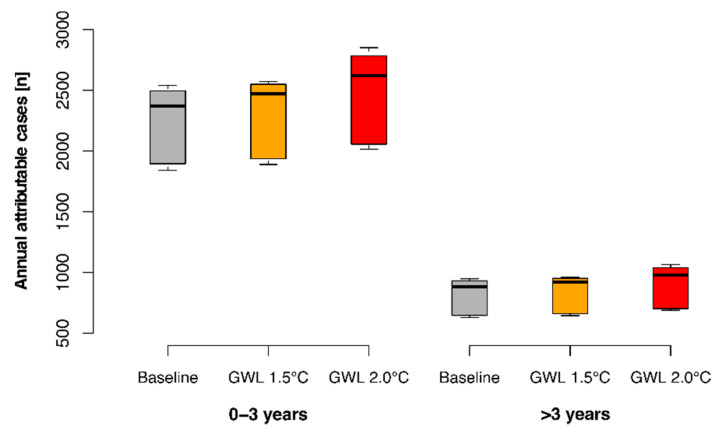
Annual number of diarrhoea cases attributable to heat (temperature > 19 °C) under current and future climate for age groups 0–3 years and >3 years.

**Figure 6 ijerph-19-04898-f006:**
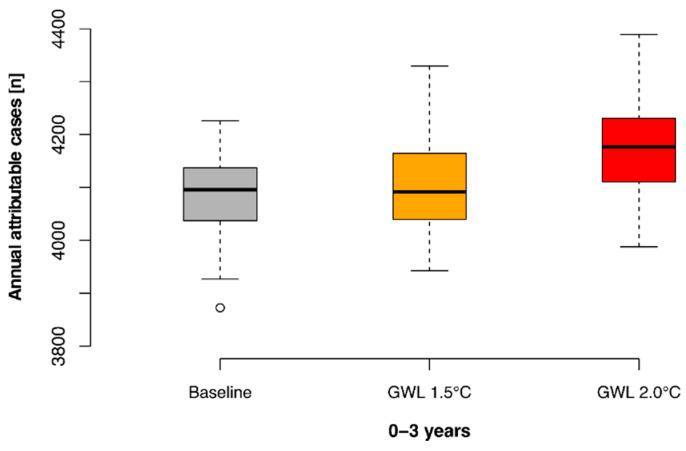
Annual number of diarrhoea cases attributable to low rainfall (<6 mm) under current and future climate for age group 0–3 years.

**Table 1 ijerph-19-04898-t001:** Overview of the climate model ensemble applied. For each model, the combination of driving Global Climate Model (GCM) and downscaling Regional Climate Model (RCM) are shown. Additionally, the 20-year periods of emergence of the 1.5 °C and 2 °C Global Warming Level (GWL) are provided.

	GCM	RCM	Spatial Resolution	Temporal Resolution	Scenario	1.5° GWL	2.0° GWL
1	CNRM-CM5	CCLM4	0.11°	daily	RCP 4.5	2028–2047	2049–2068
2	CNRM-CM5	RCA4	0.11°	daily	RCP 4.5	2028–2047	2049–2068
3	EC-EARTH	RACMO22	0.11°	daily	RCP 4.5	2013–2032	2035–2054
4	EC-EARTH	CCLM4	0.11°	daily	RCP 4.5	2013–2032	2035–2054
5	EC-EARTH	RCA4	0.11°	daily	RCP 4.5	2013–2032	2035–2054
6	IPSL-CMA5A-MR	RCA4	0.11°	daily	RCP 4.5	2007–2026	2024–2043
7	HadGEM2-ES	CCLM4	0.11°	daily	RCP 4.5	2020–2039	2036–2055
8	HadGEM2-ES	RACMO22	0.11°	daily	RCP 4.5	2020–2039	2036–2055
9	HadGEM2-ES	RCA4	0.11°	daily	RCP 4.5	2020–2039	2036–2055
10	MPI-ESM-LR	CCLM4	0.11°	daily	RCP 4.5	2013–2032	2036–2055
11	MPI-ESM-LR	RCA4	0.11°	daily	RCP 4.5	2013–2032	2036–2055
12	MPI-ESM-LR	REMO_r1	0.11°	daily	RCP 4.5	2013–2032	2036–2055
13	MPI-ESM-LR	REMO_r2	0.11°	daily	RCP 4.5	2013–2032	2036–2055

**Table 2 ijerph-19-04898-t002:** Thresholds, RR (95% CI) and annual attributable numbers of diarrhoea presentations associated with temperature or rainfall.

	Temperature (Lag 0–1 Weeks)				Rainfall (Lag 0–4 Weeks)			
Age group:	Threshold	RR per 1 °C rise above 19 °C (95% CI)	*p*-value	Annual attributable cases	Threshold	RR per 1 mm fall below 6 mm (95% CI)	*p*-value	Annual attributable cases
0–3 years	19 °C	1.008 (1.004, 1.013)	0.001	2209.0	6 mm	1.018 (1.008, 1.028)	0.001	4070.3
>3 years	19 °C	1.005 (1.001, 1.010)	0.02	797.2	6 mm	1.003 (0.994, 1.012)	0.5	n/a

## Data Availability

The data that support the findings of this study are available from the corresponding author upon reasonable request.

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
