# Peer review of "Climate Change and Diarrhoeal Disease Burdens in the Gaza Strip, Palestine: Health Impacts of 1.5 °C and 2 °C Global Warming Scenarios"

_ijerph, 2022, doi:10.3390/ijerph19084898_

Round 1

Reviewer 1 Report

Article Review: Climate change and diarrhoeal disease burdens in the Gaza Strip, Palestine: health impacts of 1.5°C and 2°C warming scenarios

The work has an important approach to a sensitive issue of public health in a region that deserves this kind of attention.

I have some suggestions and some questions:

Topic 2.1:

- I recommend that you provide us with a figure with the geographical location of the Gaza Strip.

Topic 2.3:

- I recommend showing in a Table, the basic metadata of the used RCMs.

- Has the Gaza Strip never had any surface weather stations? If not, wouldn't it be possible to get data from any station, even if not located in the Gaza strip specifically, but the closest to it, and show a basic climatology comparison of that station with the Era5-Land estimate? If possible, it will enrich your work and this result can be shown in the initial topic of results.

- Why did you decide to set future average periods where the average temperature increase will be centered at 1.5°C and 2°C? Why these fixed limits? Wouldn't it be better to analyze these future periods in all their data continuity up to 2100, even showing results for possible scenarios where the average temperature exceeds 2°C?

Topic 2.4.1:

- I suggest detailing mathematically and explaining a little better the methodologies used: Quasi-Poisson time-series regression models, Cubic-spline functions and mainly distributed lag non-linear models (DLNMs). In this only descriptive, textual way, we do not understand so well how such techniques work.

Topic 2.4.2:

- Were specific emission scenarios such as RCPs used? From what future scenarios, exactly, did you get the precipitation and temperature time series for the selected periods that centered temperature increases at 1.5°C and 2°C? This is a crucial factor to make us better understand the results presented in topics 3.2 and 3.3.

Topics 3 and 4

- Results and Discussions, are well described and the discussion is well supported. However, I also draw the attention of the authors to the way they insert the references in the text, which is not in the standard of mdpi articles, please review this.

Topic 5:

- This topic needs to be completely rewritten. In addition to being very short, it does not bring any relevant conclusion to what was researched. This topic should be based on objectively answering whether the research hypotheses were reached, and what their main results and recommendations were.

Author Response

Thank-you for the reviewer feedback on our paper. We address each of the comments in turn below and highlight where changes have been made in the revised manuscript. 

Reviewer 1

Article Review: Climate change and diarrhoeal disease burdens in the Gaza Strip, Palestine: health impacts of 1.5°C and 2°C warming scenarios

The work has an important approach to a sensitive issue of public health in a region that deserves this kind of attention.

I have some suggestions and some questions:

Topic 2.1:

- I recommend that you provide us with a figure with the geographical location of the Gaza Strip.

This has been added and is now figure 1.

Topic 2.3:

- I recommend showing in a Table, the basic metadata of the used RCMs.

Thank-you for the suggestion. This has been added and is now Table 1. 

- Has the Gaza Strip never had any surface weather stations? If not, wouldn't it be possible to get data from any station, even if not located in the Gaza strip specifically, but the closest to it, and show a basic climatology comparison of that station with the Era5-Land estimate? If possible, it will enrich your work and this result can be shown in the initial topic of results.

Yes, although there are some historical meteorological data there have been no active monitoring stations on the ground in Gaza for over 15 years. The Palestinian Meteorological Department in Gaza closed in 2007. We did consider gridded observational datasets, however, to the best of our knowledge data from E-OBS and similar only contain measurements interpolated from relatively distant weather stations from surrounding countries, so these or individual weather stations would not be representative. We have now added text in the methods to explain this and justify our exposure data source (lines 116-118).  

- Why did you decide to set future average periods where the average temperature increase will be centered at 1.5°C and 2°C? Why these fixed limits? Wouldn't it be better to analyze these future periods in all their data continuity up to 2100, even showing results for possible scenarios where the average temperature exceeds 2°C?

We chose these limits to tie-in with the IJERPH special issue on health impacts associated with such Global Warming Levels. The approach also considerably reduces uncertainty related to the projection horizon. Some models for example are more sensitive to increased CO2 concentrations than others, leading to distortion in the projected temperatures. This is particularly important for near-future time horizons. In that sense, a comparison of RCMs for earlier periods does not provide a range of uncertainty based on scenario and models alone but also adds the temporal scale. The concept of a common warming level, therefore, reduces the uncertainty and allows for a more meaningful interpretation of the outcomes, i.e., the model spread now represents a range of future outcomes that we can consider. We also believe that the application of GWLs is more relevant for policymakers. Since the 2.0°C and 1.5°C targets were the first to be articulated in the Paris agreement in 2015, by estimating the differential health impacts associated with these two targets provides greater motivation for policy action. Furthermore, uncertainties originating from different scenarios (e.g. RCP 4.5 vs. 8.5) are eliminated as fixed temperature levels are selected. We justify the choice in the introduction and this latter point is now made on lines 144-147.

Topic 2.4.1:

- I suggest detailing mathematically and explaining a little better the methodologies used: Quasi-Poisson time-series regression models, Cubic-spline functions and mainly distributed lag non-linear models (DLNMs). In this only descriptive, textual way, we do not understand so well how such techniques work.

More information on these techniques has been added to the methods section on pages 5 and 6 and the final model is now summarised in equation form.

Topic 2.4.2:

- Were specific emission scenarios such as RCPs used? From what future scenarios, exactly, did you get the precipitation and temperature time series for the selected periods that centered temperature increases at 1.5°C and 2°C? This is a crucial factor to make us better understand the results presented in topics 3.2 and 3.3.

All applied climate models follow RCP 4.5 (this has now been made clear in the new Table 1). However, as stated in the response to the warming levels above, the selection of the scenario does not influence the results of this study. For example, the selection of RCP 8.5 instead of RCP 4.5, would only result in a shift in the time period when the warming levels occur (due to higher CO2 concentrations they would occur earlier). For the meteorological variables applied in this study (temperature and precipitation), the resulting scenario uncertainty at the GWLs is considerably smaller than the model uncertainty (i.e. the model spread).

Topics 3 and 4

- Results and Discussions, are well described and the discussion is well supported. However, I also draw the attention of the authors to the way they insert the references in the text, which is not in the standard of mdpi articles, please review this.

This has been changed.

Topic 5:

- This topic needs to be completely rewritten. In addition to being very short, it does not bring any relevant conclusion to what was researched. This topic should be based on objectively answering whether the research hypotheses were reached, and what their main results and recommendations were.

The conclusion has been rewritten and expanded and now focuses more on the study findings.

Reviewer 2 Report

This is a very interesting study and newsworthy given the context.

However, some things could be improved. Firstly, how you move from the observed baseline to the model baseline and then the projections is unclear. There are some indications in the results but this is critical because it underpins the two projections of disease burden. This needs to be made much more transparent in Section 2.3. Did you use all 13 members in the ensemble, the ensemble mean or something else? Do you have a direct comparison of the model baseline for 2009-2020? It is mentioned later in passing as a small bias, but if you had a comparison of case numbers between the two that would quantify the difference, potentially increasing confidence in the results. Ok, I wrote that and had a closer look at Figure 3. This does need explanation - it shouldn't be hidden. You need to make it clear that the estimated changes are between the grey dots and the orange and red dots - however, these differences need to be established here, not snuck in later. 

It is also unclear as to whether the whole timeseries 2013-47 and 2024-2068 have been applied or averages from those. The difference between the two would indicate whether variability is represented or not. (Again this is in Fig 3 but should be made clear here)

2.4.1 The method seems sound, and the presentation of the base case is in 3.1, but the description there is inadequate. The reader needs to be sure the methods are sound and subsequent studies may draw on this but there is too little information with Figure 1. Typical is not ok - you need to show the average and the time span over which it has been aggregated. Is there a way you can show the annual variations, also (with some information about temperature and rainfall, say weeks above 19°C and below 6 mm rainfall) 

Figure 2 needs more information on the axes, such as units. These are 'clever' but not easy to interpret.

 Line 206 from model baseline levels?

Lines 216-217 Do you mean there is a rainfall bias in the models? As per above, this should be established earlier when the climate data and how it is applied is described.

Is it possible to add the observed reference to Figures 4 and 5?

Discussion

There is a lot that comes in here, some of which should be in the introduction. It will help the reader if the context is elaborated somewhat. This study proceeds as if no adaptation will occur. While it can't be predicted, this should be made clear at the start. The information on access to clean water on lines 256-58 should be in the introduction because this may also be a factor in exposure to disease that could change over time. If you take on my suggestion to show annual observed disease burden, temp and rainfall thresholds, then any underlying influence of water availability can be shown to be present or not.

The +2°C scenario is especially important because it assumes that there is no change in the underlying exposure (positive or negative) up to 2068. What should be the message here? Is it to implement mitigation goals or to show that adaptation is urgently needed? In a way, emphasising mitigation implies nothing is going to change in Gaza for that whole time.

Lines 280-287. You should be able to reinterpret this. Models that have already achieved +1.5°C within the baseline period will predict little change in disease burden, but others have not. Prudent risk management applying the precautionary principle should allow for this.   

Conclusions

Rather than indigenous populations, this should concentrate on displaced and incarcerated populations who have limited capacity to adapt. It means that even if the situation in Gaza were to improve, any population in similar circumstances would be similarly exposed.     

Author Response

Thank-you for the reviewer feedback on our paper. We address each of the comments in turn below and highlight where changes have been made in the revised manuscript. 

Reviewer 2

This is a very interesting study and newsworthy given the context.

However, some things could be improved. Firstly, how you move from the observed baseline to the model baseline and then the projections is unclear. There are some indications in the results but this is critical because it underpins the two projections of disease burden. This needs to be made much more transparent in Section 2.3. Did you use all 13 members in the ensemble, the ensemble mean or something else? Do you have a direct comparison of the model baseline for 2009-2020? It is mentioned later in passing as a small bias, but if you had a comparison of case numbers between the two that would quantify the difference, potentially increasing confidence in the results. Ok, I wrote that and had a closer look at Figure 3. This does need explanation - it shouldn't be hidden. You need to make it clear that the estimated changes are between the grey dots and the orange and red dots - however, these differences need to be established here, not snuck in later. 

Correlations between `observed‘ and model baseline were high (correlation coefficient > 0.9 for all models), however given the lack of weather station data in Gaza (please see response to reviewer 1 comments) the original baseline is also modelled and so doesn’t provide a gold standard.  We have added more text to further compare the two baselines on lines 242-249. The methods now includes more discussion of the climate data.  

Section 2.3:

It is also unclear as to whether the whole timeseries 2013-47 and 2024-2068 have been applied or averages from those. The difference between the two would indicate whether variability is represented or not. (Again this is in Fig 3 but should be made clear here)

Apologies, we have added a sentence in the methods to make this clearer (lines 193-194) and the new Table 1 describes the timespans for each climate model.  

2.4.1 The method seems sound, and the presentation of the base case is in 3.1, but the description there is inadequate. The reader needs to be sure the methods are sound and subsequent studies may draw on this but there is too little information with Figure 1. Typical is not ok - you need to show the average and the time span over which it has been aggregated. Is there a way you can show the annual variations, also (with some information about temperature and rainfall, say weeks above 19°C and below 6 mm rainfall) 

More details have been added to the methods. Figure 1 (now Figure 2) is now more fully described in the title. It is not possible to show a time-series of the whole study period as the data would be too squashed to demonstrate the seasonality each year. We have added broken lines to the figure to represent the threshold values and have indicated the number of weeks per year above/below the thresholds in the text (line 205-207).   

 Figure 2 needs more information on the axes, such as units. These are 'clever' but not easy to interpret.

This has been added.

Line 206 from model baseline levels?

Lines 216-217 Do you mean there is a rainfall bias in the models? As per above, this should be established earlier when the climate data and how it is applied is described.

Is it possible to add the observed reference to Figures 4 and 5?

Yes, it is from modelled baseline, now clarified on line 271. We avoid having two baseline estimates in the figures in order to avoid confusion. All comparisons are with the modelled baseline so that like is being compared with like. 

Discussion

There is a lot that comes in here, some of which should be in the introduction. It will help the reader if the context is elaborated somewhat. This study proceeds as if no adaptation will occur. While it can't be predicted, this should be made clear at the start. The information on access to clean water on lines 256-58 should be in the introduction because this may also be a factor in exposure to disease that could change over time. If you take on my suggestion to show annual observed disease burden, temp and rainfall thresholds, then any underlying influence of water availability can be shown to be present or not.

The +2°C scenario is especially important because it assumes that there is no change in the underlying exposure (positive or negative) up to 2068. What should be the message here? Is it to implement mitigation goals or to show that adaptation is urgently needed? In a way, emphasising mitigation implies nothing is going to change in Gaza for that whole time.

Lines 280-287. You should be able to reinterpret this. Models that have already achieved +1.5°C within the baseline period will predict little change in disease burden, but others have not. Prudent risk management applying the precautionary principle should allow for this.   

Thank-you for the suggestions. We have moved some of the discussion text into the introduction section and have added the following text into the methods: “The risk function is assumed to remain unchanged in future periods, indicating no reduced risk in future due to adaptation or, conversely, no increased vulnerability. This may be an unrealistic scenario, but our aim was to characterise the potential changes associated with climate factors only.” We agree that adaptation will be a necessary complement to mitigation and have added text in the discussion to emphasise this (lines 312-315).

Conclusions

Rather than indigenous populations, this should concentrate on displaced and incarcerated populations who have limited capacity to adapt. It means that even if the situation in Gaza were to improve, any population in similar circumstances would be similarly exposed.     

Good point, we have amended the text in the conclusions. Thank-you.

Round 2

Reviewer 1 Report

Dear authors, my questions were answered and most of the suggestions added to the text, and those that were not added, were well justified.

I recommend that in the final formatting, the subtopics are numbered, as in section 2 for example.